# KNG: The K-Norm Gradient Mechanism

**Matthew Reimherr** *
Department of Statistics
Pennsylvania State University
State College, PA 16802
mreimherr@psu.edu

**Jordan Awan**
Department of Statistics
Pennsylvania State University
State College, PA 16802
awan@psu.edu

## Abstract

This paper presents a new mechanism for producing sanitized statistical summaries that achieve *differential privacy*, called the *K-Norm Gradient* Mechanism, or KNG. This new approach maintains the strong flexibility of the exponential mechanism, while achieving the powerful utility performance of objective perturbation. KNG starts with an inherent objective function (often an empirical risk), and promotes summaries that are close to minimizing the objective by weighting according to how far the gradient of the objective function is from zero. Working with the gradient instead of the original objective function allows for additional flexibility as one can penalize using different norms. We show that, unlike the exponential mechanism, the noise added by KNG is asymptotically negligible compared to the statistical error for many problems. In addition to theoretical guarantees on privacy and utility, we confirm the utility of KNG empirically in the settings of linear and quantile regression through simulations.

## 1 Introduction

The last decade has seen a tremendous increase in research activity related to data privacy [Aggarwal and Philip, 2008, Lane et al., 2014, Machanavajjhala and Kifer, 2015, Dwork et al., 2017]. This drive has been fueled by an increasing societal concern over the large amounts of data being collected by companies, governments, and scientists. These data often contain vast amounts of personal information, for example DNA sequences, images, voice recordings, electronic health records, and internet usage patterns. Such data allows for great scientific progress by researchers and governments, as well as increasingly curated business strategies by companies. However, the such data also comes with increased risk for privacy breaches, placing greater pressure on institutions to prevent disclosures.

Currently, *Differential Privacy* (DP) [Dwork et al., 2006] is the leading framework for formally quantifying privacy risk. One of the most popular methods for achieving DP is the *Exponential Mechanism*, introduced by McSherry and Talwar [2007], and used in [Friedman and Schuster, 2010, Wasserman and Zhou, 2010, Blum et al., 2013, Dwork and Roth, 2014]. A major attribute of the exponential mechanism that contributes to its popularity is its flexibility; it can be readily adapted and incorporated into most statistical analyses. In particular, its structure makes it amenable to a wide array of statistical and machine learning problems that are based on minimizing an objective function, so called "$m$-estimators" [van der Vaart, 2000, Chapter 5]. Some examples where the exponential mechanism has been used include PCA [Chaudhuri et al., 2013, Awan et al., 2019], hypothesis testing [Canonne et al., 2019], maximum likelihood estimation (related to posterior sampling) [Wang et al., 2015, Minami et al., 2016], and density estimation [Wasserman and Zhou, 2010].

However, examples have arisen [Wang et al., 2015, Awan et al., 2019] where the magnitude of the noise added by the exponential mechanism is substantially higher than other mechanisms. Recently,

Awan et al. [2019], demonstrated that, in a very broad sense, the exponential mechanism adds noise that is not asymptotically negligible relative to the statistical estimation error, which other mechanism are able to achieve in different problems [e.g. Smith, 2011]. In this paper we provide a new mechanism called the *K-Norm Gradient Mechanism*, or *KNG*, that retains the flexibility of the exponential mechanism, but with substantially improved utility guarantees. KNG provides a principled approach to developing efficient mechanisms that also perform well in practice. Indeed the Laplace, $K$-norm, and PrivateQuantile mechanisms can all be viewed as instantiations of KNG. Here we also use KNG to provide the first mechanism for private quantile regression that we are aware of, which we empirically show is efficient.

At a high level, KNG uses a similar perspective to that of the exponential mechanism. In particular, suppose that $\ell_n(\theta; D)$ is an objective, whose minimizer, $\hat{\theta} \in \mathbb{R}^d$, is the summary we aim to sanitize. Here $D$ represents the particular database and $n$ the sample size of $D$. The exponential mechanism aims to release $\tilde{\theta}_E$ based on the density

$$f_E(\theta) \propto \exp\{-c_0 \ell_n(\theta; D)\},$$

where $c_0$ is a generic constant determined by the sensitivity of $\ell_n$ and the desired level of privacy. Conceptually, the idea is to promote sanitized estimates whose utility, as measured by $\ell_n$, is close to that of $\hat{\theta}$. Unfortunately, Awan et al. [2019], showed that the magnitude of the noise added by the exponential mechanism is often of the same order as the statistical error (as a function of $n$), resulting in inefficient private estimators. KNG uses a similar perspective, but takes the gradient of $\ell_n$ and promotes $\theta$ that are close to the solution $\nabla \ell_n(\hat{\theta}) = 0$. Since we work with the gradient, we also have the flexibility of choosing a desirable norm, which Awan and Slavković [2018] showed can be tailored to the problem at hand to achieve better utility. The resulting mechanism produces a sanitized $\tilde{\theta}$ according to the density

$$f_n(\theta) \propto \exp\{-c_0 \|\nabla \ell_n(\theta; D)\|_K\},$$

where $\|\cdot\|_K$ is a general norm on $\mathbb{R}^d$ that can be chosen to accommodate the context of the problem. Here we see a connection between KNG and the *$K$-norm mechanism*, introduced by Hardt and Talwar [2010]. The terminology is based on the idea of considering a set $K$ which is the convex hull of the *sensitivity polytope* [Kattis and Nikolov, 2017], and defining $\|\cdot\|_K$ to be the norm such that the ball of radius one is $K$, i.e. $\{v \in \mathbb{R}^d \mid \|v\|_K = 1\} = K$. In fact every norm can be generated in this manner, so no there is no loss in generality from using this approach [Awan and Slavković, 2018].

KNG can similarly be viewed as a modification of objective perturbation [Chaudhuri et al., 2011, Kifer et al., 2012]. There, one releases a sanitized estimate, $\tilde{\theta}_O$, by minimizing[2]

$$\tilde{\theta}_O = \mathrm{argmin}_{\theta \in \Theta} \left( \ell_n(\theta; D) + \omega \theta^\top b \right),$$

where $b \in \mathbb{R}^d$ is a random vector with distribution drawn from the $K$-norm mechanism $f_b(x) \propto \exp\{-\|b\|_K\}$, and $\omega \in \mathbb{R}$ is a fixed constant based on the sensitivity of $\ell_n$ and the desired level of privacy[3]. Equivalently, one has that $\nabla \ell_n(\tilde{\theta}_O; D) + \omega b = 0$, which implies that $\tilde{\theta}_O = \nabla \ell_n^{-1}(-\omega b)$, assuming $\nabla \ell_n$ is invertible. Using the change of variables formula, this implies that $\tilde{\theta}_O$ has density

$$f_O(\theta) \propto \exp\{-\omega^{-1} \|\nabla \ell_n(\theta)\|_K\} |\det(\nabla^2 \ell_n(\theta))|.$$

With KNG, the second derivative term $\nabla^2 \ell_n$ is not included. Furthermore, there are several technical requirements when working with objective perturbation that KNG sidesteps. In particular, the proof that objective perturbation satisfies DP requires the objective function to be strongly convex and twice differentiable almost everywhere [Chaudhuri et al., 2011, Kifer et al., 2012, Awan and Slavković, 2018]. While we assume strong convexity and a second derivative to prove a utility result in Theorem 3.2, KNG does not require either of these conditions to satisfy DP. This allows the KNG mechanism to be applied in more general situations (such as median estimation and quantile regression, explored in Section 4), and requires fewer calculations to implement.

The remainder of this paper is organized as follows. In Section 2 we recall the necessary background on differential privacy and the exponential mechanism. In Section 3 we formally define KNG and

show that it achieves $\epsilon$-DP with nearly the same flexibility as the exponential mechanism. We also provide a general utility result that shows that the noise introduced by KNG is of order $O_p(n^{-1})$, which is negligible compared to the statistical estimation error, which is typically $O_p(n^{-1/2})$. We also show that the noise introduced by KNG is asymptotically from a $K$-norm mechanism. In section 4 we provide several examples of KNG applied to statistical problems, including mean estimation, linear regression, median/quantile estimation, and quantile regression. We also illustrate the empirical advantages of KNG in the settings of linear and quantile regression through simulations. We conclude in Section 5 by discussing challenges and potential extensions of KNG.

## 2 Differential Privacy Background

Differential privacy (DP), introduced by Dwork et al. [2006] has taken hold as the primary framework for formally quantifying privacy risk. Several versions of DP have been proposed, such as approximate DP [Dwork and Roth, 2014], concentrated DP [Dwork and Rothblum, 2016, Bun and Steinke, 2016], and local DP [Duchi et al., 2013], all of which fit into the axiomatic treatment of formal privacy given by Kifer and Lin [2012]. In this paper, we work with pure $\epsilon$-DP, stated in Definition 2.1.

Let $\mathcal{D}^n$ denote the collection of all possible databases with $n$ units. The bivariate function $\delta : \mathcal{D}^n \times \mathcal{D}^n \to \mathbb{R}$, which maps $\delta(D, D') := \#\{i \mid D_i \neq D_i'\}$, is called the *Hamming Distance* on $\mathcal{D}^n$. It is easy to verify that $\delta$ is a metric on $\mathcal{D}^n$. If $\delta(D, D') = 1$ then $D$ and $D'$ are said to be *adjacent*.

Let $f : \mathcal{D}^n \to \Theta$ represent a summary of $\mathcal{D}^n$, and $\mathcal{F}$ a $\sigma$-algebra on $\Theta$, such that $(\Theta, \mathcal{F})$ is a measurable space. A *privacy mechanism* is a family of probability measures $\{\mu_D : D \in \mathcal{D}^n\}$ over $\Theta$.

**Definition 2.1** (Differential Privacy: Dwork et al., 2006). A privacy mechanism $\{\mu_D : D \in \mathcal{D}^n\}$ satisfies $\epsilon$-Differential Privacy ($\epsilon$-DP) if for all $B \in \mathcal{F}$ and adjacent $D, D' \in \mathcal{D}^n$,

$$\mu_D(B) \leq \mu_{D'}(B) \exp(\epsilon).$$

The exponential mechanism, introduced by McSherry and Talwar [2007] is a central tool in the design of DP mechanisms [Dwork and Roth, 2014]. In fact every mechanism can be viewed as an instance of the exponential mechanism, by setting the objective function as the log-density of the mechanism. In practice, it is most common to set the objective as a natural loss function, such as an empirical risk.

**Proposition 2.2** (Exponential Mechanism: McSherry and Talwar, 2007). *Let $(\Theta, \mathcal{F}, \nu)$ be a measure space, and let $\{\ell_n(\theta; D) : \Theta \to \mathbb{R} \mid D \in \mathcal{D}^n\}$ be a collection of measurable functions indexed by the database $D$. We say that this collection has a finite sensitivity $\Delta$, if*

$$|\ell_n(\theta; D) - \ell_n(\theta; D')| \leq \Delta < \infty,$$

*for all adjacent $D, D'$ and $\nu$-almost all $\theta \in \Theta$. If $\int_\Theta \exp(-\ell_n(\theta; D)) \, d\nu(\theta) < \infty$ for all $D \in \mathcal{D}$, then the collection of probability measures $\{\mu_D \mid D \in \mathcal{D}\}$ with densities (with respect to $\nu$)*

$$f_D(\theta) \propto \exp\left\{ \left( \frac{-\epsilon}{2\Delta} \right) \ell_n(\theta; D) \right\} \quad \textit{satisfies } \epsilon\textit{-DP.}$$

Intuitively, $\ell_n(\theta; D)$ provides a score quantifying the utility of an output $\theta$ for the database $D$. We use the convention that smaller values of $\ell_n(\theta; D)$ provide more utility. So, the exponential mechanism places more mass near the minimizers of $\ell$, and less mass the higher the value of $\ell_n(\theta; D)$.

## 3 The K-Norm Gradient Mechanism

In Section 2 we considered an arbitrary measure space, $(\theta, \mathcal{F}, \nu)$, when defining DP and the exponential mechanism. However, here we focus on $\mathbb{R}^d$. The KNG mechanism cannot be defined to quite the generality of the exponential mechanism since we require enough structure on the parameter space to define a gradient. Most applications focus on Euclidean spaces, so this is not a major practical concern, but there could be implications for more complicated nonlinear, discrete, or infinite dimensional settings.

**Theorem 3.1** ($K$-Norm Gradient Mechanism (KNG)). *Let $\Theta \subset \mathbb{R}^d$ be a convex set, $\|\cdot\|_K$ be a norm on $\mathbb{R}^d$, and $\nu$ be a $\sigma$-finite measure on $\Theta$. Let $\{\ell_n(\theta; D) : \Theta \to \mathbb{R} \mid D \in \mathcal{D}^n\}$ be a collection of*

*measurable functions, which are differentiable $\nu$ almost everywhere. We say that this collection has sensitivity $\Delta : \Theta \to \mathbb{R}^+$, if*

$$\|\nabla \ell_n(\theta; D) - \nabla \ell_n(\theta; D')\|_K \leq \Delta(\theta) < \infty,$$

*for all adjacent $D, D'$ and $\nu$-almost all $\theta$. If $\int_{\Theta} \exp(-\frac{1}{\Delta(\theta)}\|\nabla \ell_n(\theta; D)\|_K) \, d\nu(\theta) < \infty$ for all $D \in \mathcal{D}$, then the collection of probability measures $\{\mu_D \mid D \in \mathcal{D}\}$ with densities (with respect to $\nu$)*

$$f_D(\theta) \propto \exp\left[\left(\frac{-\epsilon}{2\Delta(\theta)}\right)\|\nabla \ell_n(\theta; D)\|_K\right] \quad \textit{satisfies } \epsilon\text{-DP.}$$

*Proof.* Set $\widetilde{\ell}_n(\theta; D) = \Delta(\theta)^{-1}\|\nabla \ell_n(\theta; D)\|_K$. Then $\widetilde{\ell}$ has sensitivity 1. By Proposition 2.2, the described mechanism satisfies $\epsilon$-DP. □

One advantage of this approach over the traditional exponential mechanism is that the sensitivity calculation is often simpler (e.g. quantile regression, subsection 4.5). However, it also has the same intuition as the exponential mechanism. In particular, the optimum, $\hat{\theta}$, occurs when $\nabla \ell_n(\hat{\theta}) = 0$, thus we want to promote solutions that make the gradient close to 0, and discourage ones that make the gradient far from 0. These concepts are closely related to $m$-estimators, $z$-estimators, and estimating equations [van der Vaart, 2000, Chapter 5].

Since KNG utilizes the gradient, it links in nicely to optimization methods such as gradient descent. However, it could also suffer from some of the same challenges as gradient descent. Namely, if the objective function has multiple local minima, then KNG will promote output near each these points. For this reason, a great deal of care should be taken with KNG when applying to non-convex objective functions, such as fitting neural networks [Gori and Tesi, 1992].

## 3.1 Asymptotic Properties

While flexibility of a mechanism is an important concern, ultimately the utility of the output is of primary importance. Awan et al. [2019] showed that for a large class of objective functions, the exponential mechanism introduces noise of magnitude $O_p(n^{-1/2})$, where $n$ is the sample size. For many statistical problems the non-private error rate is also $O_p(n^{-1/2})$ [van der Vaart, 2000, Chapter 5], meaning that the exponential mechanism introduces noise that is not asymptotically negligible.

Under similar assumptions, we show in Theorem 3.2 that KNG has aymptotic error $O_p(n^{-1})$, which is asymptotically negligible compared to the statistical error. In fact, Theorem 3.2 shows that the noise introduced is asymptotically from a $K$-norm mechanism [Hardt and Talwar, 2010, Awan and Slavković, 2018], which generalizes the Laplace mechanism.

The assumptions in Theorem 3.2 are chosen to capture a large class of common loss functions, which include many convex empirical risk functions and log-likelihood functions. Mathematically, the assumption that $\ell$ is twice-differentiable and strongly convex allow us to use a one term Taylor expansion of $\nabla \ell$ about $\hat{\theta}$, and guarantee that the integrating constants converge. The proof of Theorem 3.2 is found in the Supplementary Materials.

**Theorem 3.2** (Utility of KNG). *Let $\Theta \subset \mathbb{R}^d$ be a convex set, $\|\cdot\|_K$ a norm on $\mathbb{R}^d$, $\nu$ a $\sigma$-finite measure om $\Theta$, and $\ell_n(\theta) := \ell_n(\theta; D)$ be a sequence of objective functions which satisfy the assumptions of Theorem 3.1, with sensitivity $\Delta(\theta)$. We further assume that*

1. *$n^{-1}\ell_n(\theta)$ are twice differentiable (almost everywhere) convex functions and there exists a finite $\alpha > 0$ such that $n^{-1}\mathbf{H}_n(\theta)$ has eigenvalues greater than $\alpha$. for all $n$ and $\theta \in \Theta$;*

2. *the minimizers satisfy $\hat{\theta} \to \theta^\star \in \mathbb{R}^d$ and $n^{-1}\mathbf{H}_n(\hat{\theta}) \to \Sigma^{-1}$ where $\Sigma$ is a $d \times d$ positive definite matrix;*

3. *$\Delta(\theta)$ is continous in $\theta$, constant in $n$, and there exists $\Delta > 0$ such that $\Delta \leq \Delta(\theta)$.*

*Assume the base measure, $\nu$, has a bounded, differentiable density $g(\theta)$ (with respect to Lebesgue measure) which is strictly positive in a neighborhood of $\theta^\star$. Then the sanitized value $\tilde{\theta}$ drawn from the KNG mechanism with privacy parameter $\epsilon$ is asymptotically $K$-norm. That is, the density of*

$Z = n(\tilde{\theta} - \hat{\theta})$ *converges to a K-norm distribution, with density (wrt $\nu$) proportional to* $f(z) \propto$ $\exp\left(\frac{-\epsilon}{2\Delta(\theta^*)}\|\Sigma^{-1}z\|_K\right).$

The proof of the CLT for the exponential mechanism in Awan et al. [2019], as well as the proof of Theorem 3.2, both rely on a Taylor expansion of the objective function. In both cases, it is assumed that the Hessian converges, when scaled by $n$, to a positive definite matrix. However, using the original objective function requires two derivatives before the Hessian appears in the Taylor expansion, whereas the use of the gradient only requires one derivative. The consequence of this is that the traditional exponential mechanism results in a quadratic numerator inside the exponent, whereas KNG has a (normed) linear numerator. Asymptotically, this gives an $O_p(n^{-1/2})$ Gaussian noise for the exponential mechanism and an $O_p(n^{-1})$ $K$-norm noise for KNG. Geometrically, it seems that the use of an objective function which behaves linearly (in absolute value) near the optimum, rather than quadratic, results in better asymptotic utility. By using the normed-gradient, we construct an objective function with this property.

The assumptions in Theorem 3.2 are very similar to the assumptions for the CLT in Awan et al. [2019]. So, whenever these properties hold, we know that KNG results in an $O_p(n^{-1})$ privacy noise whereas the exponential mechanism is $O_p(n^{-1/2})$. To further emphasize the importance of this result, we note that the magnitude of the noise introduced for privacy can have a substantial impact on the sample complexity. Asymptotically, KNG requires exactly the same sample size as the non-private estimator, whereas the exponential mechanism requires a constant $> 1$ multiple of the non-private sample size to achieve the same accuracy.

As we see in Section 4, in the problem of quantile regression the assumptions of Theorem 3.2 do not hold, meaning that while we guarantee privacy in that setting, we can't guarantee the utility of the estimator. However, we see in Figure 2 that KNG still introduces asymptotically negligible noise, suggesting that the assumptions of Theorem 3.2 can likely be weakened to accomodate a larger class of objective functions.

**Remark 3.3.** Based on the discussion in Section 1, a result similar to 3.2 may hold for objective perturbation as well. The main issue is dealing with the change of variables factor $|\det \mathbf{H_n}(\theta)|$, which may or may not contribute to the asymptotic form. We suspect that when both KNG and objective perturbation are applicable (e.g. linear regression, see subsection 4.3), they will have similar performance. However, as KNG does not require a second derivative (or convexity), it is applicable in more settings than objective perturbation (e.g. quantile regression, see subsection 4.5).

## 4 Examples

### 4.1 Mean Estimation

Mean estimation is one of the simplest statistical tasks, and one of the first to be solved in DP. Assuming bounds on the data, the mean can be estimated by adding Laplace noise [Dwork et al., 2006]. Recently there has been some work developing statistical tools for the mean under differential privacy, such as confidence intervals in the normal model [Karwa and Vadhan, 2017] and hypothesis tests for Bernouilli data [Awan and Slavković, 2018]. We show that KNG recovers the $K$-norm mechanism when estimating the mean, a generalization of the Laplace mechanism.

Let $x_1, \ldots, x_n \in \mathbb{R}^d$, which we assume are drawn from some population with mean $\theta^*$. To estimate $\theta^*$, we use the sum of squares as our objective function:

$$\ell_n(\theta; D) = \sum_{i=1}^n \|x_i - \theta\|_2^2 \quad \text{and} \quad \nabla\ell_n(\theta; D) = -2\sum_{i=1}^n (x_i - \theta) = -2n(\bar{x} - \theta).$$

Turning to the sensitivity, if we assume that there exists a constant $r$ such that $\|x_i\|_K \leq r < \infty$ for some norm $\|\cdot\|_K$, then the sensitivity of the gradient is $\|\nabla\ell_n(\theta; D) - \nabla\ell_n(\theta; D')\|_K = 2\|x_1 - x_1'\|_K \leq 2r$. Thus the mechanism becomes $f_n(\theta) \propto \exp\{-(n\epsilon/(4r))\|\bar{x} - \theta\|_K\}$, which is exactly a K-norm mechanism [Hardt and Talwar, 2010]. So $\tilde{\theta} - \bar{x}$ has mean 0 and standard deviation $O_p(n^{-1})$. Thus, the noise added for privacy is asymptotically negligible compared to the statistical error $O_p(n^{-1/2})$.

**Remark 4.1.** Because the KNG results in a location family in this case, the integrating constant does not depend on the data. So, we do not need to divide $\epsilon$ by 2 in the density, and may instead draw from $f_n(\theta) \propto \exp\left\{ \frac{n\epsilon}{2r} \|\bar{x} - \theta\|_K \right\}$, which is how the $K$-norm mechanism is normally stated.

## 4.2 Linear Regression

There has been a great deal of work developing DP methods for linear regression [Zhang et al., 2012, Song et al., 2013, Dwork and Lei, 2009, Chaudhuri et al., 2011, Kifer et al., 2012, Sheffet, 2017]. In this section, we detail how KNG can be used to estimate the coefficients in a linear regression model. We observe pairs of data $(x_i, y_i)$, where $y_i \in \mathbb{R}$ and $x_i \in \mathbb{R}^d$, which we assume are modeled as $y_i = x_i^\top \theta^* + e_i$, where the errors are iid with mean zero and are uncorrelated with $x$. Our goal is to estimate $\theta^*$. To implement KNG, we assume that the data has been pre-processed such that $-1 \leq x_i \leq 1$ and $-1 \leq y_i \leq 1$ for all $i = 1, \ldots, n$. We also assume that $\|\theta^*\|_1 \leq B$. The usual non-private estimator for $\theta^*$ is the least-squares, which minimizes the objective function $\ell(\theta; D) = \sum_{i=1}^n (y_i - x_i^\top \theta)^2$. KNG requires a bound on the sensitivity of $\nabla \ell_n$:

$$\|\nabla \ell_n(\theta; D) - \nabla \ell_n(\theta; D')\| \leq \sup_{y_1, x_1, \theta} 4\|(y_1 - x_1^\top \theta)x_1\| = \sup_{x_1} 4(1+B)\|x_1\|.$$

By using the $\ell_\infty$ norm, we get the tightest bound, since $\|x_1\|_\infty \leq 1$. KNG samples from the density

$$f_n(\theta) \propto \exp\left( \frac{-\epsilon}{8(1+B)} \left\| \sum_{i=1}^n (y_i - x_i^\top \theta)x_1^\top \right\|_\infty \right), \tag{1}$$

with respect to the uniform measure on $\Theta = \{\theta \mid \|\theta\|_1 \leq B\}$.

**Remark 4.2.** Alternative sensitivity bounds can be obtained by choosing other bounds on $x$ and $y$. The bound on $\theta^*$ can be removed entirely, allowing $\Delta$ to depend on $\theta$. In that case, a nontrivial base measure will be required as the resulting density is not integrable with respect to Lebesgue measure. We prefer to use the given sensitivity bound as it allows a fairer comparison against the exponential mechanism and objective perturbation in subsection 4.3.

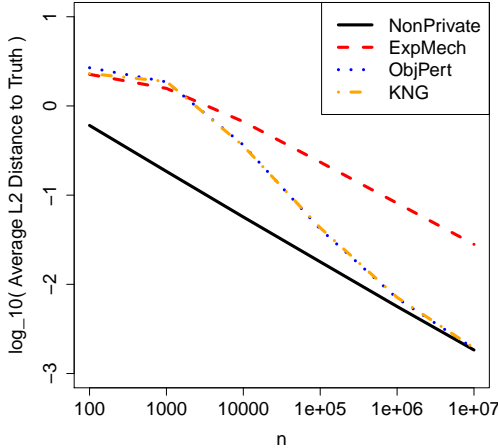

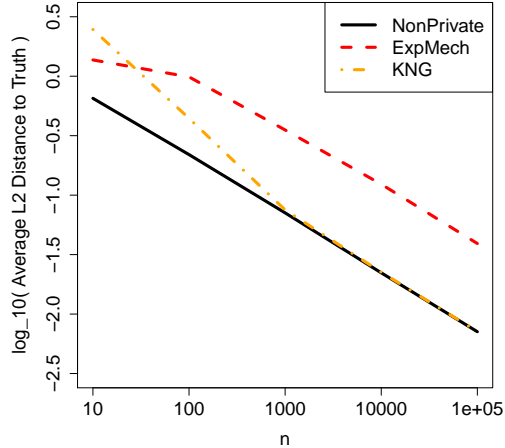

Figure 1: Simulation comparing the non-private MLE, exponential mechanism, objective perturbation, and KNG for linear regression.

Figure 2: Simulation comparing the non-private, exponential mechanism, and KNG for quantile regression.

## 4.3 Linear Regression Simulation

In this section, we examine the finite sample performance of the KNG mechanism on linear regression compared to the exponential mechanism and objective perturbation mechanism. KNG samples from the density (1), the exponential mechanism samples from

$$f_n(\theta) \propto \exp\left( \frac{-\epsilon}{2(1+B)^2} \sum_{i=1}^n (y_i - x_i^\top \theta)^2 \right),$$

and objective perturbation draws a random vector $b$ from the density $f(b) \propto \exp\left(-\frac{\epsilon}{8(1+B)}\|b\|_\infty\right)$, and then finds the optimum of the modified objective: $\arg\min_{\|\theta\|_1 \leq 1} \ell_n(\theta; D) + \frac{\gamma}{2}\theta^\top\theta + \theta^\top b$, where $\gamma = (\exp(\epsilon/2) - 1)^{-1}(2d)$ and $d$ is the dimension of the $x_i$'s. For all three mechanisms we assume the bound on $\|\theta^*\|_1$ is $B = 1$. Details on these mechanisms for linear regression can be found in the Supplementary Materials.

For the simulations the true regression vector $\theta^* \in \mathbb{R}^{12}$ is $\theta^* = (0, -1, -1+2/11, -1+4/11, \ldots, 1-2/11)$, and so $d = 12$. For each $n$ in $10^2, 10^3, 10^4, \ldots, 10^7$ we run 100 replicates of Algorithm 1 at $\epsilon = 1$. For KNG and exponential mechanism, we draw samples using a one-at-a-time MCMC procedure with 10000 steps.

At the end, we compute the average distance over the 100 replicates for each mechanism and for each sample size $n$. The results are plotted in Figure 1, taking the base 10 log of both axes. At each $n$ value and for each mechanism, the Monte Carlo standard errors are between 0.01380 and 0.02729, in terms of the log-scale used in the plot. The benefit of plotting in this fashion is that it makes it easier to understand the asymptotic behavior of each estimator.

Since we know that the estimation error of the non-private MLE is error $= Cn^{-1/2}$, taking the log of both sides shows that the convergence should appear as a straight line with slope $-1/2$: $\log(\text{error}) = -\frac{1}{2}\log(n) + \log(C)$, which is the black line in Figure 1.

As Awan et al. [2019] showed, the asymptotic estimation error of the exponential mechanism is error $= Kn^{-1/2}$, where $K$ is a constant greater than $C$. Taking the log of both sides gives another line with slope $-1/2$, but with a higher intercept: $\log(\text{error}) = -\frac{1}{2}\log(n) + \log(K)$, which we see in red in Figure 1.

On the other hand, for KNG and objective perturbation (based on Remark 3.3), the asymptotic estimation error is error $= Cn^{-1/2} + Kn^{-1}$, which when logged shows that for larger $n$, the curve approaches the line of the non-private estimation error from above: $\log(\text{error}) = -\frac{1}{2}\log(n) + \log(C + Kn^{-1/2})$, which is also confirmed in Figure 1.

---

**Algorithm 1** Regression Simulation

---

INPUT: $n, \epsilon, d, \theta^*$.

1: Generate $X \in \mathbb{R}^{n \times d}$ such that $X_{i,1} = 1$ and $X_{ij} \overset{\text{iid}}{\sim} U(-1, 1)$ for $i = 1, \ldots, n$ and $j = 2, \ldots, d$.
2: Generate independent errors $e_i \sim N(0, 1)$ for $i = 1, \ldots, n$.
3: Compute the responses $Y_i = X_i\theta^* + e_i$.
4: Set $R = \max_i |Y_i|$.
5: Set $Y_i' = Y_i/R$.
6: Use $X$ and $Y'$ to estimate the regression coefficient via the non-private estimator, and each DP mechanism.
7: Multiply the estimates by $R$ to estimate $\theta^*$.
8: Compute the euclidean distance between the estimate and the true $\theta^*$ for each estimator.

OUTPUT: Average distances of the estimates to the true $\theta^*$.

---

## 4.4 Median Estimation

Just as in the mean estimation problem, we observe $D = (x_1, \ldots, x_n)$, where $x_i \in \mathbb{R}^d$, and our goal is to estimate the population median. In the case when $d = 1$, the median can be estimated using the empirical risk function $\ell_n(\theta; D) = \sum_{i=1}^n |x_i - \theta|$. In general for $d \geq 1$, we are estimating the *geometric median* [Minsker et al., 2015], which can be expressed as $\arg\min_m \mathbb{E}\|X - m\|$, and typically the euclidean norm is used. Now, our objective becomes $\ell_n(\theta; D) = \sum_{i=1}^n \|x_i - \theta\|$. It may be concerning that this objective is not differentiable everywhere, however, KNG only requires that the gradient exist on a set of measure one. The gradient of $\|x_i - \theta\|$ in our norm's topology is given by $d(\theta, x_i) := \|x_i - \theta\|^{-1}(x_i - \theta)$, provided that $\theta \neq x_i$. Notice that this gives a direction in $\mathbb{R}^d$ since $\|d(\theta, x_i)\| = 1$. Using the triangle inequality, we see that the sensitivity of the gradient is bounded by 2. So the KNG mechanism for the median can be expressed as

$$f_n(\theta) \propto \exp\left\{-\frac{\epsilon n}{4}\left\|\frac{1}{n}\sum_{i=1}^n d(\theta, x_i)\right\|\right\}.$$

Again, the error introduced is $O_p(n^{-1})$, which is negligible compared to the statistical error.

## 4.5 Quantile Regression

For quantile regression as for linear regression, we observe pairs of data $(x_i, y_i)$, where $y_i \in \mathbb{R}$ and $x_i \in \mathbb{R}^d$. We assume that $Q_{Y_i|X_i}(\tau) = X_i^\top \theta_\tau^*$, for all $i = 1, \ldots, n$, where $Q_{Y|X}(\tau)$ is the conditional quantile function of $Y$ given $X$ for $0 < \tau < 1$, and $\theta^* \in \mathbb{R}^p$ [Hao et al., 2007]. For a given $\tau$, $\theta_\tau^*$ can be estimated as $\hat{\theta}_\tau = \arg\min_\theta \sum_{i=1}^n \rho_\tau(y_i - x_i^\top \theta)$, where $\rho_\tau(z) = (\tau-1)zI(z \le 0) + \tau z I(z > 0)$ is called the *tiled absolute value function* [Koenker and Hallock, 2001]. So, our objective function is

$$\ell_n(\theta; D) = (\tau - 1) \sum_{y_i \le x_i^\top \theta} (y_i - x_i^\top \theta) + \tau \sum_{y_i > x_i^\top \theta} (y_i - x_i^\top \theta),$$

with gradient (almost everywhere)

$$\nabla \ell_n(\theta; D) = (\tau - 1) \sum_{y_i \le x_i^\top \theta} (-x_i) + \tau \sum_{y_i > x_i^\top \theta} (-x_i) = -\tau \sum_{i=1}^n x_i + \sum_{y_i \le x_i^\top \theta} x_i.$$

We bound the sensitivity as $\Delta = 2(1-\tau)C_X$, where $\sup_{x_1} \|x_1\| \le C_X$. Then KNG samples from

$$f_n(\theta) \propto \exp\left\{ \frac{-\epsilon n}{4(1-\tau)C_X} \left\| -\tau \frac{1}{n} \sum_{i=1}^n x_i + \frac{1}{n} \sum_{y_i \le x_i^\top \theta} x_i \right\| \right\}. \tag{2}$$

We see a few nice benefits of the KNG method in this example. If we were to use $\ell_n$ directly in the exponential mechanism, then not only would we expect worse asymptotic performance (as demonstrated in subsection 4.5.1), but we see that the sensitivity calculation for the gradient only requires a bound on $X$, whereas the sensitivity of $\ell_n$ requires bounds on $Y$, $X$, and $\theta^*$. Furthermore, the objective perturbation mechanism cannot be used in this setting, because $\ell$ is not strongly convex, whereas the proofs for objective perturbation [Chaudhuri and Monteleoni, 2009, Chaudhuri et al., 2011, Kifer et al., 2012, Awan and Slavković, 2018] all require strong convexity. In fact, the Hessian of $\ell_n$ is zero almost everywhere making the objective perturbation inapplicable.

Finally note that if we are only interested in estimating the $\tau^{th}$ quantile of a set of real numbers $Y_1, \ldots, Y_n$, we could set $X_i = 1$ for all $i = 1, \ldots, n$, in which case KNG samples from

$$f_n(\theta) \propto \exp\left\{ \frac{-\epsilon n}{4(1-\tau)} \left| \tau - \hat{F}(\theta; Y) \right| \right\}. \tag{3}$$

In fact, this is the *Private Quantile* algorithm proposed by Smith [2011], who also establish strong utility guarantees for the algorithm; this exercise demonstrates that KNG could provide, or at least contribute to, a more unified framework for developing efficient privacy mechanisms.

### 4.5.1 Quantile Regression Simulation

In this section, we examine the empirical performance of the KNG mechanism on quantile regression compared to the exponential mechanism. KNG samples from the density (2) using the $\|\cdot\|_\infty$ norm and setting $C_X = 1$, and the exponential mechanism samples from

$$f_n(\theta) \propto \exp\left\{ \frac{-\epsilon}{4\max\{\tau, 1-\tau\}(1+B)} \ell_n(\theta; D) \right\}.$$

We assume, as in subsection 4.3 that $B = 1$. Details on the exponential mechanism can be found in the Supplementary Materials. Note that objective perturbation cannot be used in this setting, as discussed in subsection 4.5.

For the simulations, we use $\tau = 1/2$ and the true regression vector $\theta_{1/2}^* \in \mathbb{R}^2$ is $\theta_{1/2}^* = (0, -1)$. For each $n$ in $10^1, 10^2, \ldots, 10^5$ we run 100 replicates of Algorithm 1 at $\epsilon = 1$. Samples from KNG and the exponential mechanism are obtained using 1000 steps of a one-at-a-time MCMC algorithm. At the end, we compute the average distance over the 100 replicates for each estimator and for each sample size $n$. The results are plotted in Figure 1, taking the base 10 log of both axes. At each $n$ value and for each mechanism, the monte carlo standard errors are between 0.04403 and 0.06028, in terms of the log-scale.

We see in figure 2 that the non-private estimate appears as a straight line with slope $-1/2$, reflecting the fact that its estimation error is $O_p(n^{-1/2})$. We also see that the exponential mechanism approaches a line with slope $-1/2$, but with a higher intercept, reflecting that it has increased asymptotic variance. Last, we see that the error of KNG approaches the error line of the non-private estimator, suggesting that KNG has the same asymptotic rate as the non-private estimator.

While the utility guarantees of Theorem 3.2 do not apply in this setting, as the objective function is not strongly convex, the santized estimates still achieve $\epsilon$-DP and we see from Figure 2 that, empirically, KNG introduces $o_p(n^{-1/2})$ error in this setting as well. This suggests that the assumptions in Theorem 3.2 can likely be weakened, and KNG in fact produces efficient mechanisms for an even broader set of problems than Theorem 3.2 prescribes.

## 5  Conclusions

In this paper we presented a new privacy mechanism, KNG, that maintains much of the flexbility of the exponential mechanism, while having substantially better utility guarantees. These guarantees are similar to those provided by objective perturbation, but privacy can be achieved with far fewer structural assumptions. A major draw back of the mechanism is the same as for gradient descent, which can have trouble with local minima or saddle points. Two interesting open questions concern the finite sample efficiency of KNG vs objective perturbation and if KNG can be adapted or combined with other methods to better handle multiple minima.

We also believe that KNG has a great deal of potential for handling infinite dimensional and nonlinear problems. For example, parameter spaces consisting of Hilbert spaces or Riemannian manifolds have structures that allow for the computation of gradients, and which might be amenable to KNG. With Riemannian manifolds, the gradient is often viewed as a linear mapping over tangent spaces, while in Hilbert spaces, the gradient is often treated as a linear functional. A major advantage of KNG over other mechanisms is the direct incorporation of a general $K$-norm. Awan et al. [2019] showed that the exponential mechanism has major problems over function spaces, which are of interest in nonparametric statistics. These issues could potentially be alleviated by KNG with a careful choice of norm. Many interesting challenges remain in data privacy, especially if there is additional complicated structure in the parameters or data.

KNG has strong connections with prior DP mechanisms, especially the exponential mechanism and objective perturbation. Indeed, like nearly every privacy mechanism, KNG can be phrased as very particular type of exponential mechanism, however this doesn't provide insight into why KNG achieves better statistical properties. In particular, a key point is to consider the objective function that motivated the original statistical summary, which, when used with KNG produces sanitized estimators with better statistical performance than the classic implementation of the exponential mechanism.

One downside of KNG is the issue of sampling, which is similar to the exponential mechanism in that sampling from these distributions is, in general, non-trivial. We show that for mean and quantile estimation, KNG results in distributions that are efficiently sampled. However, for linear and quantile regression, we used a one-at-a-time MCMC procedure (also used for exponential mechanism). Just like sampling from an posterior distribution, developing a convenient sampling scheme is case-by-case, but often a simple MCMC procedure works well in practice.

## Acknowledgements

This research was supported in part by NSF DMS 1712826, NSF SES 1853209, and NSF SES-153443 to The Pennsylvania State University. The first author is also grateful for the hospitality of the Simons Institute for the Theory of Computing at UC Berkeley.

## Footnotes

*Research supported in part by NSF DMS 1712826, NSF SES 1853209, and the Simons Institute for the Theory of Computing at UC Berkeley.

[2]In fact, objective perturbation minimizes $\ell_n(\theta; D) + c\theta^\top \theta + \omega \theta^\top b$, where $c$ is a constant. We ignore this regularization term in this discussion for the simplicity of the illustration.

[3]In Chaudhuri et al. [2011] and Kifer et al. [2012], the $\ell_2$ norm is used. Awan and Slavković [2018] extend objective perturbation to allow for arbitrary norms.

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
