[Supplementary Material · KNG_SUPPLEMENT.pdf]

# KNG: The K-Norm Gradient Mechanism: Supplementary Materials

**Matthew Reimherr**
Department of Statistics
Pennsylvania State University
State College, PA 16802
mreimherr@psu.edu

**Jordan Awan**
Department of Statistics
Pennsylvania State University
State College, PA 16802
awan@psu.edu

## 1 Proofs

*Proof of Theorem 3.2.* For notational simplicity, we assume that the base measure, $\mu$, is Lebesgue. The density of the KNG mechanism can then be expressed as

$$f_n(\theta) = c_n^{-1} \exp \left\{ \frac{-\epsilon}{2\Delta(\hat{\theta} + z/n)} \|\nabla \ell_n(\theta)\|_K \right\},$$

where $c_n$ is the normalizing constant. Define the random variable $Z = n(\tilde{\theta} - \hat{\theta})$, then its density is given by

$$f_n(z) = c_n^{-1} n^{-1} \exp \left\{ \frac{-\epsilon}{2\Delta(\hat{\theta} + z/n)} \|\nabla \ell_n(\hat{\theta} + z/n)\|_K \right\}.$$

Using a one term Taylor expansion, we have by Assumption (2) and (3) that

$$\nabla \ell_n(\hat{\theta} + z/n) = \nabla \ell_n(\hat{\theta}) + \mathbf{H}_n(\hat{\theta})z/n + o_p(1)$$
$$= \mathbf{H}_n(\hat{\theta})z/n + o_p(1),$$

where $\mathbf{H}_n(\theta)$ is the Hessian matrix of $\ell_n$ evaluated at $\theta$. Recall that

$$c_n n = \int \exp \left\{ \frac{\epsilon}{2\Delta(\hat{\theta} + z/n)} \left( -\|\nabla \ell_n(\hat{\theta} + z/n)\|_K \right) \right\} dz.$$

By Assumption (1), $\ell_n$ is strongly convex and thus

$$\frac{1}{\Delta(\hat{\theta} + z/n)} \left\langle \nabla \ell_n(\hat{\theta} + z/n) - \nabla \ell_n(\hat{\theta}), z/n \right\rangle \geq \frac{n\alpha}{\Delta} \|z/n\|_2^2.$$

Combining the Cauchy-Schwartz inequality with the fact that $\nabla \ell_n(\hat{\theta}) = 0$ implies

$$\frac{1}{\Delta(\hat{\theta} + z/n)} \|\nabla \ell_n(\hat{\theta} + z/n)\|_2 \geq \frac{n\alpha}{\Delta} \|z/n\|_2.$$

By the equivalence of norms on $\mathbb{R}^d$, we have that

$$\frac{-1}{\Delta(\hat{\theta} + z/n)} \|\nabla \ell_n(\hat{\theta} + z/n)\|_K \leq \frac{-C\alpha}{\Delta} \|z\|_2,$$

for some constant $C$. Since $\exp\{-\|z\|_2\}$ is integrable, we can apply the dominated convergence theorem to conclude that the constants converge to a nonzero and finite quantity. Since $\Delta(\theta)$ is

continuous in $\theta$, we also have that $\Delta(\hat{\theta} + z/n) \to \Delta(\theta^*)$. Putting everything together, we can conclude that

$$f_n(z) \to f(z) \propto \exp\left\{\frac{-\epsilon}{2\Delta(\theta^*)}\|\Sigma^{-1}z\|_K\right\},$$

which is the density of the $K$-norm mechanism. Applying Scheffe's Theorem, we thus have both convergence in distribution as well as convergence in total variation to a $K$-norm mechanism $\qquad\square$

## 2 Linear Regression

### 2.1 Exponential Mechanism

Our objective function is $\ell(\theta; D) = \sum_{i=1}^n (y_i - x_i^\top \theta)^2$. For the exponential mechanism, we need to bound the sensitivity of $\ell(\theta)$:

$$\begin{aligned}
|\ell_n(\theta; D) - \ell_n(\theta; D')| &= |(y_1 - x_1^\top \theta)^2 - (y_2 - x_2^\top \theta)^2| \\
&\leq \sup_{y_1, x_1, \theta} (y_1 - x_1^\top \theta)^2 \\
&\leq \sup_{x_1, \theta} (1 + |x_1^\top \theta|)^2 \\
&\leq \sup_{\theta} (1 + \|\theta\|_1)^2 \\
&= (1 + B)^2,
\end{aligned}$$

where we used the assumptions that $\|x_1\|_\infty \leq 1$, $|y_1| \leq 1$, and $\|\theta^*\|_1 \leq B$. The exponential mechanism with objective function $\ell(\theta)$ draws $\theta$ from

$$f_n(\theta) \propto \exp\left(\frac{-\epsilon}{2(1+B)^2}\sum_{i=1}^n (y_i - x_i^\top \theta)^2\right),$$

with respect to the uniform measure on $\{\theta \mid \|\theta\|_1 \leq B\}$.

### 2.2 Objective Perturbation

For objective perturbation, we use the version stated in Awan and Slavković [2018], which allows us to use the same bound on the gradient as developed in subsection 4.2. Objective perturbation also requires a bound on the eigenvalues of the hessian for one datapoint:

$$\begin{aligned}
\text{max eigenvalue}(2x_1 x_1^\top) &\leq \text{trace}(2x_1 x_1^\top) \\
&= 2\,\text{trace}(x_1^\top x_1) \\
&\leq 2\sum_{j=1}^d |x_{1j}|^2 \\
&\leq 2d.
\end{aligned}$$

Objective perturbation then draws a random vector $b$ from the density $f(b) \propto \exp\left(-\frac{\epsilon}{8(1+B)}\|b\|_\infty\right)$ (a simple sampling algorithm for $f(b)$ is stated in Awan and Slavković [2018]), and then finds the optimum of the modified objective:

$$\arg\min_{\|\theta\|_1 \leq 1} \ell_n(\theta; D) + \frac{\gamma}{2}\theta^\top \theta + \theta^\top b,$$

where $\gamma = \frac{2d}{\exp(\epsilon/2)-1}$. Since $\ell$ is convex, this new objective is also convex. We restrict the search space of $\theta$ to $\{\theta \mid \|\theta\|_1 \leq B\}$, since we assume that $\|\theta^*\| \leq B$ for our bounds.

# 3 Quantile Regression

## 3.1 Exponential Mechanism

Our objective function is $\ell_n(\theta; D) = \sum_{i=1}^{n} \rho_\tau(y_i - x_i^\top \beta)$. For the exponential mechanism we need to assume additional bounds on the data as well as on $\theta^*$. As in the linear regression case, we assume that $-1 \leq y_i \leq 1$, $-1 \leq x_i \leq 1$, and $\|\theta\|_1 \leq B$. We bound the sensitivity of $\ell_n$ as

$$
\begin{aligned}
|\ell_n(\theta; D) - \ell_n(\theta; D')| &\leq 2\,|\ell_n(\theta; D)| \\
&\leq \sup_{y_1, x_1, \theta} 2\max\{\tau, 1 - \tau\}\,\big|y_1 - x_1^\top \theta\big| \\
&\leq 2\max\{\tau, 1 - \tau\}(1 + B).
\end{aligned}
$$

The exponential mechanism then samples from the density

$$
f_n(\theta) \propto \exp\left\{\frac{-\epsilon}{4\max\{\tau, 1 - \tau\}(1 + B)}\ell_n(\theta; D)\right\}.
$$

# References

Jordan Awan and Aleksandra Slavković. Structure and sensitivity in differential privacy: Comparing $k$-norm mechanisms. *ArXiv e-prints*, January 2018. Under Review.