[Reviews · NeurIPS 2019]

Reviewer 1



I find the results of this paper interesting both in the theoretical bounds they allow to achieve and in the practical applicability of the proposed mechanism to multiple classical statistical problems. Even if the main result (Theorem 3.2) is rather simple, it has important consequences both theoretically and experimentally, as shown in the rest of the paper. I find also interesting to see that the considered class of functions allow one to go beyond what the general exponential mechanism can achieve. Finally, it is interesting to see the relations between this method and objetive perturbation. One aspect that I find questionable in this work is the claim that it is introducing a new mechanism. The proposed mechanism can be seen as a particular instance of the exponential mechanism with a weight function based on the gradient of the objective function, rather than a new mechanism per se. Pros: -the paper identifies an interesting class of weighting functions for the exponential mechanism. -the proposed method can be applied to a quite large class of examples. Cons: -calling the proposed mechanism a "new" mechanism is questionable. Comments after rebuttal ---------------------------- Thanks for your answer. Concerning the "new" mechanism, I don't want to open up a controversy since this is really a minor point but some of the paper you mention, e.g. the K-norm paper, acknowledge that their mechanism are an instance of the exponential mechanism when they introduce them. I am not claiming that you should not give it a new name. I don't think that your contribution would be weaker by acknowledging this. More importantly, I encourage you to include the more technical discussion on the comparison with the exponential mechanism and the lower bound by Awan et al. in the paper.

Reviewer 2



This work presents and motivates the KNG mechanism, which stands somewhat as hybrid of the exponential mechanism and objective perturbation. It relies somewhat on assumptions about the task, but with much simpler assumptions than those needed by objective perturbation. This proposed method is also grounded in terms of an existing K-norm mechanism. Originality: the proposed KNG and its theoretical properties are novel. Quality: This work is exceptionally complete, with theory, empirical studies, case study applications, and relations to other approaches. Clarity: the work is clear and structured Significance: The work provides a significant algorithm, with compelling asymptotic behaviors on tasks without requiring as onerous assumptions as other methods.

Reviewer 3



I thank the authors for shedding some light on the separation between exponential mechanism and the proposed K-norm mechanism based on the lower bounds from Awan et al. ICML 2019. Please include a proper discussion on this topic in the updated version. ------- The paper introduces the K-norm gradient (KNG) mechanism as a way of achieving differential privacy. It is similar in spirit to the exponential mechanism; with the key difference is that the loss function in the exponential sampling is replaced by an appropriate norm of the gradient of the loss function. The authors analyze the asymptotic error of this KNG mechanism under some assumptions on the loss function, and argue that it could be better that the asymptotic error of the exponential mechanism in some cases. The paper also discusses specific instantiation of this KNG mechanism on statistical problems such as mean estimation, linear regression, and quantile estimation/regression. The results look correct and are well-presented. The paper however does not seem to have a standout technical idea or contribution. The results feel incremental and for a theoretical paper lacks the required technical novelty expected for NeurIPS.

[Author Response · NeurIPS 2019]

We thank the reviewers for their interest, and hope this document clarifies our unique contributions.
In the final version, we will include additional exposition addressing each of the following points.

Reviewer #2 questions whether it is appropriate to describe our method as a new mechanism, since
it can be viewed as a particular case of the exponential mechanism. In fact many widely used DP
mechanisms can be expressed as an instance of the exponential mechanism, such as the Laplace
mechanism, geometric mechanism, staircase mechanism, K-norm mechanism, posterior sampling
mechanism, etc. As we show, the KNG approach is applicable in a wide variety of situations, and
offers improved utility over a classic implementation of the exponential mechanism. For these reasons,
we think it is justified to refer to KNG as its own mechanism.

Reviewer #3 asks for intuition behind the use of the gradient in KNG. The proof of the CLT for the
exponential mechanism in Awan et al. 2019 (ICML), as well as the proof of Theorem 3.2, both rely
on a Taylor expansion of the objective function. In both cases, it is assumed that the hessian converges
at a $O(n)$ rate to a positive definite matrix. However, using the original objective function requires
two derivatives before the Hessian appears in the Taylor expansion, whereas the use of the gradient
only requires one derivative. The consequence of this is that the traditional exponential mechanism
results in a quadratic numerator, whereas KNG has a (normed) linear numerator. Asymptotically, this
gives $O(1/\sqrt{n})$ Gaussian noise for the exponential mechanism and $O(1/n)$ $K$-norm noise for KNG.

Geometrically, it seems that the use of an objective function which behaves linearly (in absolute
value) near the optimum, rather than quadratic, results in better asymptotic utility. By using the
normed-gradient, we construct an objective function with this property.

Reviewer #4 asks for clarification on how the performance of KNG differs from the exponential
mechanism. While it may not have been clear in the exposition of our submission, in fact the
assumptions in THM 3.2 are nearly identical to the assumptions required in the CLT of Awan et al.
2019 (ICML). So, for any problem in which these assumptions are satisfied, KNG always results in
$O(1/n)$ noise, whereas exponential mechanism results in $O(1/\sqrt{n})$ noise.

Reviewer #4 also asks for interesting problems where KNG outperforms the exponential mechanism.
Among the examples in the manuscript, mean estimation and linear regression both satisfy all of the
assumptions to justify that KNG results in $O(1/n)$ noise, whereas exponential mechanism results in
$O(1/\sqrt{n})$ noise. While the problems of median estimation and quantile regression do not satisfy the
assumptions of THM 3.2 (though they are still private), we demonstrated empirically via simulations
that KNG still results in $O(1/n)$ whereas exponential mechanism results in $O(1/\sqrt{n})$ noise.

To emphasize the improvement that KNG offers over the exponential mechanism, we point out
that adding $O(1/n)$ versus $O(1/\sqrt{n})$ noise has a substantial impact on the sample complexity.
Asymptotically, KNG requires exactly the same sample size as the non-private estimator, whereas
exponential mechanism requires a constant >1 multiple of the non-private sample size.

We also note that THM 3.2, as well as the examples of median estimation and quantile regression,
indicate that KNG outperforms the exponential mechanism for a wide variety of interesting problems.
Many log-likelihoods fit this framework, as well as many other empirical risk functions.

Reviewer #4 also asks about sampling algorithms for the KNG mechanisms. KNG is similar to the
exponential mechanism in that sampling these distributions is generally non-trivial. We show that for
mean and quantile estimation, KNG results in distributions that are efficiently sampled. However, for
linear and quantile regression, we used a one-at-a-time MCMC procedure (also used for exponential
mechanism). Just like sampling from an posterior distribution, developing a convenient sampling
scheme is case-by-case, but often a simple MCMC procedure works well in practice.

Finally, Reviewer #4 questions whether the results in this manuscript are significant enough for
publication in NeurIPS. While we acknowledge that the proof of THM 3.2 is not technically com-
plex, we argue that KNG offers both an important theoretical and practical contribution to the DP
literature. While it has been shown before that asymptotically efficient mechanisms exist [Smith
2011], constructing practical and efficient mechanisms for a particular problem is non-trivial. KNG
offers a principled approach to developing efficient mechanisms. For mean and quantile estimation,
KNG offers a method of constructing both the Laplace and PrivateQuantile mechanisms. However,
besides unifying these prior mechanisms, KNG can also be used to build mechanisms for problems
not previously solved. In fact, using KNG we develop the first DP mechanism for quantile regression
that we are aware of, and demonstrate empirically that the mechanism is asymptotically efficient.



[Meta-Review · NeurIPS 2019]

The authors propose a new differentially private mechanism for a large class of statistical tasks. The reviewers thought the paper was interesting, and the fact that the new mechanism gets around deficiencies of directly applying the exponential mechanism in certain settings is significant.